# MOMENTUM CONSERVING LAGRANGIAN NEURAL NETWORKS

## ABSTRACT

Realistic models of physical world rely on differentiable symmetries that, in turn, correspond to conservation laws. Recent works on *Lagrangian* and *Hamiltonian* neural networks show that the underlying symmetries of a system can be easily learned by a neural network when provided with an appropriate inductive bias. However, these models still suffer from issues such as inability to generalize to arbitrary system sizes, poor interpretability, and most importantly, inability to learn translational and rotational symmetries, which lead to the conservation laws of linear and angular momentum, respectively. Here, we present a *momentum conserving Lagrangian neural network (*MCLNN*)* that learns the Lagrangian of a system, while also preserving the translational and rotational symmetries. We test our approach on linear and non-linear spring systems, and a gravitational system, demonstrating the energy and momentum conservation. We also show that the model developed can generalize to systems of any arbitrary size. Finally, we discuss the interpretability of the MCLNN, which directly provides physical insights into the interactions of multi-particle systems.

## 1 INTRODUCTION AND RELATED WORK

Realistic modeling of the time-evolution of multi-particle physical systems lie at the heart of several fields of science and engineering (Goldstein, 2011; Park et al., 2021; Roehrl et al., 2020; Lutter et al., 2019). Examples include multiple bodies connected by springs, bodies interacting under gravitational forces, or even systems at the atomic and mesoscale such as proteins or colloidal gels, which range several orders ($\approx 10^{20}$) of length and timescales from atomic systems to planetary systems (Park et al., 2021; Cranmer et al., 2020b). Traditionally, the trajectory of these systems are obtained by solving the associated differential equations numerically. These differential equations, in turn, are derived based on the invariant or conserved quantities of a system. Recent studies on *Lagrangian (*LNN) and *Hamiltonian neural networks* show that one of these invariant quantities, energy, can be learned directly from the data enabling realistic simulation of systems (Cranmer et al., 2020a; Finzi et al., 2020; Lutter et al., 2019; Greydanus et al., 2019; Zhong et al., 2021; 2019; Zhong & Leonard, 2020). However, these approaches fail to model *multi-particle* interactions effectively due to the following limitations.

• **Conservation of momentum:** Invoking Noether's theorem (Noether, 1971), *"a differentiable symmetry in action results in a conserved quantity"*. Conversely, *"every conserved quantity can be derived from an underlying symmetry in action"*. It can be shown that an interacting multi-particle system, that is closed, respects three conservation laws, namely, energy, linear momentum and angular momentum (Noether, 1971; Goldstein, 2011). While the first is a consequence of symmetry of the system with respect to time, the second and third is a consequence of symmetry with respect to translation and rotation in space, respectively. Thus, the Lagrangian of such a system remains invariant under the translation and rotation operations on the Cartesian coordinates. For example, three balls connected by a spring will have the same Lagrangian (and hence the interaction forces) when the system as a whole is subjected to rotation or translation. That is, as long as the relative positions of the balls are preserved, translation or rotation by any magnitude will not have an effect on the system dynamics. While LNNs have been demonstrated to learn the symmetry in time by conserving the energy (Cranmer et al., 2020a), they have not been designed to respect the other two conservation laws with respect to momentum. Learning these symmetries directly from the data is an extremely hard problem and may require large amounts of data. Furthermore, as we will show

in Sec. 4, even when trained on large volumes of training data, the LNNs fail to generalize well on unseen data. This questions the realistic nature of the dynamics simulated by LNNs.

• **Generalizability to unseen system sizes:** LNNs lack the ability to generalize to system sizes beyond the training set. For instance, an LNN trained on three balls connected by a spring cannot model a system of four balls connected by a spring despite having the same conserved quantities and interactions. This limits the broader applicability of LNNs to realistic physical systems.

• **Interpretability:** LNN (Cranmer et al., 2020a) directly learns (and predicts) the Lagrangian of a system as a function of the position and velocities of all the particles in the system. Owing to this design, it is not possible to recover the inter-particle dynamics within a multi-particle system in the form of pairwise potential energy or forces. This limits the interpretability of the LNN as the contribution of positions and velocities towards the total Lagrangian of the system is represented using a black-box deep neural network.

In this paper, we address the above limitations. Specifically, our contributions are as follows:

• **Problem Formulation and Architecture Design:** We reformulate LNN (Cranmer et al., 2020a) with a relational inductive bias by applying a transformation on the *Cartesian coordinates* of the system and decoupling the terms of the Lagrangian. Armed with this reformulation, we develop a *momentum conserving* LNN called MCLNN. MCLNN introduces several key innovations. **(1)** First, MCLNN preserves the rotational and translation symmetries. **(2)** Second, MCLNN generalizes to unseen system sizes. **(3)** Third, MCLNN generates interpretable models with the output characterizing pair-wise interactions of the bodies in the system, and thereby providing physical insights into the system dynamics.

• **Theoretical Characterization:** We rigorously establish that MCLNN conserves energy, linear and angular momentum. Furthermore, in contrast to LNN (Cranmer et al., 2020a), we show that MCLNN can directly learn trajectories (positions) of systems without requiring training on acceleration. This is a desirable property. In many cases including experimental systems, access to acceleration of each entity at every time instant may not be possible. While position of particles is a direct observable, acceleration is a derived quantity.

• **Empirical Evaluation:** We perform in-depth evaluation across three multi-particle systems namely, balls connected by linear and non-linear springs, and gravitational system. Our experiments establish that MCLNN performs significantly better than LNN with long-term stability, generalizes to arbitrary-sized multi-particle systems, and is capable of learning interaction dynamics with 3-4 orders of magnitude lower volume of training data.

## 2 PRELIMINARIES AND PROBLEM FORMULATION

In this section, we introduce the preliminary concepts central to our work and formulate the problem. As notational convention, vectors are represented with an overhead arrow (Ex. $\vec{v}$) and higher-order tensors, such as matrices, in **bold**. A summary notations used is provided in the appendix A.7.

**Definition 1 (Multi-particle System)** *An $\mathcal{N}$-body multi-particle system contains $\mathcal{N}$ particles $P = \{n_1, \cdots, n_{\mathcal{N}}\}$. At any given time $t$, particle $n_i$ is characterized by its position $\vec{q}_i^{\,t}$ and velocity $\dot{\vec{q}}_i^{\,t}$.*

The positions of particles in an $\mathcal{N}$-body system are not static. They change due to various interaction forces at play (Ex: a gravitational system). These positional changes are captured in the form of *trajectories*.

**Definition 2 (Multi-particle Trajectory)** *The trajectory of an $\mathcal{N}$-body multi-particle system $P$ over a time horizon $T = [t_s, t_e]$ is the sequence of position and velocity vectors $(\mathbf{q}^t, \dot{\mathbf{q}}^t) \mid t \in T\}$, where $\mathbf{q}^t = \{\vec{q}_i^{\,t} \mid n_i \in P\}$ and $\dot{\mathbf{q}}^t = \{\dot{\vec{q}}_i^{\,t} \mid n_i \in P\}$.*

In physics, the time-evolution or *trajectory* of interacting particles is obtained by solving the differential equations of motion. The numerical solution to these equations provide the acceleration of the particle, which can then be used to obtain the updated velocity and positions. Indeed, the acceleration of particles can be directly learned by neural networks as a function of its position and velocities. However, it fails to learn the underlying symmetries, which in turn leads to the laws of conservation of energy and momenta.

Recent approaches to predict trajectories by learning the Lagrangian through an LNN (Cranmer et al., 2020a; Finzi et al., 2020; Zhong et al., 2021; Zhong & Leonard, 2020) has been shown to be effective in learning the symmetry in time (Cranmer et al., 2020a; Zhong et al., 2021). Lagrangian, defined as $L(\mathbf{q}, \dot{\mathbf{q}}) = T(\dot{\mathbf{q}}) - V(\mathbf{q})$, is a scalar functional of kinetic energy ($T(\dot{\mathbf{q}})$) and potential energy ($V(\mathbf{q})$),

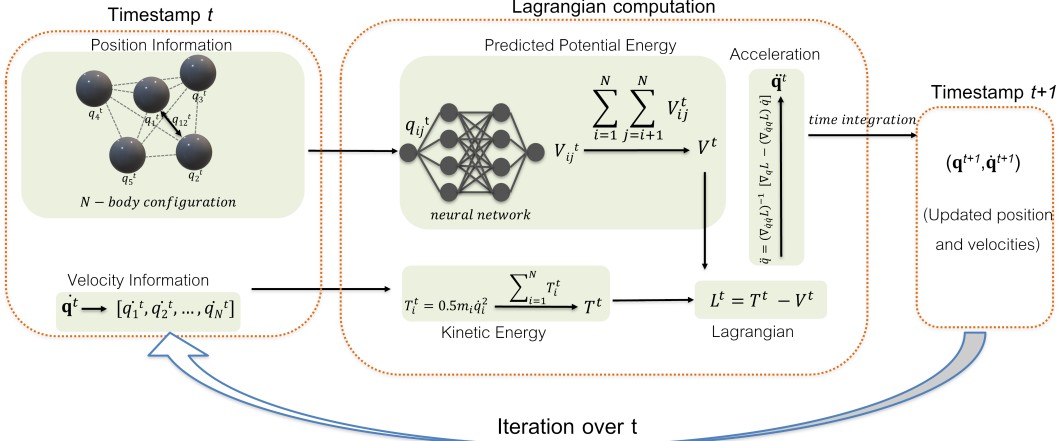

Figure 1: MCLNN framework.

that depends on both the set of positions $\mathbf{q}$, and velocities $\dot{\mathbf{q}}$ of the particles in the system (Goldstein, 2011). LNN bypasses the necessity to learn accelerations for each particle and directly predicts the scalar Lagrangian for a system. Additionally, the trajectory predicted using *Euler-Lagrange (EL)* equation conserves energy resulting in an overall better prediction of the system dynamics.

Theoretically, although learning the Lagrangian of a system is enough to predict its dynamics, the Lagrangian itself exhibits some symmetries that are not imposed by the EL equation. Lagrangian of a system exhibits translational and rotational symmetry. Specifically, the Lagrangian of a system $L(\mathbf{q}, \dot{\mathbf{q}})$ and $L(\mathbf{q}', \dot{\mathbf{q}})$, where $\mathbf{q}' = \{\vec{q_i} + \vec{\epsilon} \mid n_i \in P\}$ for any finite value of $\vec{\epsilon}$ are equivalent. Similarly, $L(\mathbf{q}, \dot{\mathbf{q}})$ and $L(\mathbf{Q}\mathbf{q}, \dot{\mathbf{q}})$, where $\mathbf{Q}$ is an *orthogonal tensor* representing a rotation, are equivalent. These symmetries are not provided as an inductive bias in the LNN framework.

Further, LNNs are trained by giving the coordinates as input and minimizing the loss between the actual acceleration and acceleration predicted by LNN through EL equations. This requires *a priori* access to the actual acceleration, which may not be available in many cases, for example, experimental trajectory of colloidal gels visualized by video camera.

Here, we aim to address these open challenges by reformulating the LNN with a relational inductive bias, resulting in *Momentum Conserving* LNN (MCLNN). Specifically, we aim to learn the dynamics of multi-particle interacting systems purely from the trajectory of the constituent particles.

**Problem 1 (Momentum Conserving LNN for Trajectory Prediction)** *Let $\mathbb{T}$ be a set of trajectories of $\mathcal{N}$-body systems. Furthermore, let there be a hidden joint distribution of physics-constrained configurational and temporal space from which $\mathbb{T}$ has been sampled. Our goal is to learn this hidden distribution. Towards that end, we want to learn:*

  *i. A generative model $p(\mathbb{T})$ that maximizes the likelihood of generating $\mathbb{T}$.*
  *ii. The generative model must respect the laws of physics such as conservation of energy and momenta.*

*Once learned, this generative model can be used to predict trajectories of unseen $\mathcal{N}$-body systems.*

## 3 THEORY

The flowchart of MCLNN is shown in Fig. 1. In this approach, the positions and velocities at any time $t$ is used to predict the trajectory of the system, while respecting the conservation laws of energy and momenta. We analyze the performance of MCLNN on three systems, namely, linear spring, non-linear spring, and gravitational system to predict the accurate dynamics of these multi-particle interacting systems.

### 3.1 EULER-LAGRANGE EQUATION

Consider a system of $\mathcal{N}$-particles interacting with each other under a potential. Let us define functional, namely, "action", $S$ as:

$$S = \int_{t_0}^{t_1} L \, dt \tag{1}$$

where $L$ is the Lagrangian of the system. Then trajectory taken by this system to move from the position $\mathbf{q^0}$ to $\mathbf{q^1}$ in time $t_0$ to $t_1$ is the one that makes the "action" $S$ stationary. This leads to the

equation of motion governing the dynamics of system, namely, the EL equation as:

$$\frac{d}{dt}\frac{\partial L}{\partial \dot{\mathbf{q}}} = \frac{\partial L}{\partial \mathbf{q}} \tag{2}$$

The acceleration of particles in the system, $\ddot{\mathbf{q}} = \{\ddot{\vec{q}}_i \mid n_i \in P\}$ can be computed directly from the EL equation (see Appendix A.3) as:

$$\ddot{\mathbf{q}} = (\nabla_{\dot{\mathbf{q}}\dot{\mathbf{q}}}L)^{-1} \left[\nabla_{\mathbf{q}}L - (\nabla_{\dot{\mathbf{q}}\mathbf{q}}L)\,\dot{\mathbf{q}}\right] \tag{3}$$

This acceleration can then be used to compute the updated positions and velocities of particles. Note that EL equation leads to the conservation of *Hamiltonian*, defined as: $H(\mathbf{q}, \dot{\mathbf{q}}) = T(\dot{\mathbf{q}}) + V(\mathbf{q})$, which also represents the *total energy* of the system.

## 3.2 MOMENTUM CONSERVING LAGRANGIAN NEURAL NETWORK

As mentioned earlier, Lagrangians exhibit translational and rotational symmetry. Learning this directly from the positions of particles is extremely challenging. To address this challenge, we reformulate the LNN. Consider five balls connected by linear springs. We will use this running example to explain MCLNN. The input for the MCLNN is the position and velocities at any time $t$. Further, we identify the set of all the pairs of particles in the system as $(i, j)$. For a given set, we compute the difference between the positions followed by computing the $l^2$ norm to obtain the pairwise distances $q_{ij}$, which corresponds to the length of the spring. Thus,

$$q_{ij} = \sqrt{(\vec{q}_i - \vec{q}_j) \cdot (\vec{q}_i - \vec{q}_j)} \tag{4}$$

The $q_{ij}$ is given as the input to a neural network which outputs a scalar $V_{ij}$. Note that the $V_{ij}$ may be considered to be the pair-wise potential energy of a spring connecting two particles $i$ and $j$. $V_{ij}$ summed over all the pairs of particles gives the $V(\mathbf{q})$, the total potential energy of the system. For multi-particle systems, the kinetic energy of a particle $n_i$ is the function of its velocity $\dot{\vec{q}}_i$, can defined as (Goldstein, 2011; Zhong et al., 2021):

$$T_i(\dot{\vec{q}}_i) = \frac{1}{2}m_i\dot{\vec{q}}_i^2. \tag{5}$$

Here, we invoke this expression of kinetic energy, which when summed over all the particles provide the total kinetic energy, $T$. Then, $T$ and $V$ are used to obtain the $L$ as in Eq. 7. The $L$ when substituted into the EL equation (Eq. 2), provides the acceleration and the updated position and velocities of the system using the velocity-Verlet integration (Rapaport, 2004). The loss function of MCLNN is on the predicted and actual positions at time $t + 1, t + 2, \ldots, t + n$, which is then back-propagated to train the neural network. Specifically, the loss function is as follows.

$$\mathcal{L} = \frac{1}{|\mathbb{T}|}\sum_{\forall \mathcal{T} \in \mathbb{T}} \left( \frac{1}{|\mathcal{T}|}\sum_{t=2}^{|\mathcal{T}|}\frac{1}{\mathcal{N}} \left(\sum_{i=1}^{\mathcal{N}}\left(\mathcal{T}.q_i^t - p\left(\mathcal{T}.q_i^t\right)\right)^2\right) \right) \tag{6}$$

Here, $p(\mathcal{T}.q_i^t)$ is the predicted position for the $i^{th}$ particle in $\mathcal{T}$ at time $t$ and $\mathcal{T}.q_i^t$ is the true position. $|\mathcal{T}|$ denotes the last time step in trajectory $\mathcal{T}$. The updated expression for Lagrangian for an $\mathcal{N}$-particle system can be written as:

$$L(\mathbf{q}, \dot{\mathbf{q}}) = \sum_{i=1}^{n}\frac{1}{2}m_i\dot{\vec{q}}_i^2 - \sum_{i=1}^{n-1}\sum_{j=i+1}^{n}V_{ij}(q_{ij}) \tag{7}$$

It should be noted that the specific form of kinetic energy $T$ does not limit the applicability of the present approach (Zhong et al., 2021). Since, the focus of the work is to model multi-particle systems and it is well-known that the kinetic energy of the system generally follows Eq. 5, we choose this expression to simplify the problem. In cases where the expression for kinetic energy is not known or is different, the function can be replaced by an additional neural network with an approach similar to that for $V$.

## 3.3 TRANSLATIONAL AND ROTATIONAL SYMMETRY OF MCLNN

**Theorem 1** MCLNN *exhibits translational and rotational symmetry.*

PROOF. Consider, the Lagrangian of the system of $\mathcal{N}$-particles as $L(\mathbf{q}, \dot{\mathbf{q}}) = T(\dot{\mathbf{q}}) - V(\mathbf{q})$. First, we focus on the translational symmetry and next on rotational symmetry.

**Lemma 1** MCLNN *exhibits translational symmetry.*

PROOF. When the particles are subjected to a translation $\vec{\epsilon}$, the updated positions are given by $\mathbf{q}' == \{\vec{q}_i + \vec{\epsilon} \mid n_i \in P\}$. The updated Lagrangian of the system is given by $L(\mathbf{q}', \dot{\mathbf{q}})$. To prove $L(\mathbf{q}', \dot{\mathbf{q}}) = L(\mathbf{q}, \dot{\mathbf{q}})$, we need to prove $V(\mathbf{q}') = V(\mathbf{q})$ as the other term in the Lagrangian remains unaffected. Applying the transformation on the positions and computing the $l^2$ norm,

$$
\begin{aligned}
q'_{ij} &= \sqrt{[(\vec{q}_i + \vec{\epsilon}) - (\vec{q}_j + \vec{\epsilon})] \cdot [(\vec{q}_i + \vec{\epsilon}) - (\vec{q}_j + \vec{\epsilon})])} \\
&= \sqrt{(\vec{q}_i - \vec{q}_j) \cdot (\vec{q}_i - \vec{q}_j)} \\
&= q_{ij}
\end{aligned}
\tag{8}
$$

which implies $V(\mathbf{q}') = V(\mathbf{q})$. Therefore, $L(\mathbf{q}', \dot{\mathbf{q}}) = L(\mathbf{q}, \dot{\mathbf{q}})$.

**Lemma 2** MCLNN *exhibits rotational symmetry.*

PROOF. When the system is subjected to pure rotation, the positions are transformed as $\mathbf{Qq}$, where $\mathbf{Q}$ is an orthogonal tensor representing a rotation. Correspondingly, the Lagrangian is modified as $L(\mathbf{Qq}, \dot{\mathbf{q}})$. As in the case of translation, we need only to prove that $V(\mathbf{Qq}) = V(\mathbf{q})$. To prove this, it is worth recalling that, for an orthogonal tensor $\mathbf{Q}$, $\mathbf{Q}\mathbf{Q}^t = \mathbf{Q}^t\mathbf{Q} = I$. Now, the proof can be obtained by computing $q'_{ij}$ as:

$$
\begin{aligned}
q'_{ij} &= \sqrt{(\mathbf{Q}\vec{q}_i - \mathbf{Q}\vec{q}_j) \cdot (\mathbf{Q}\vec{q}_i - \mathbf{Q}\vec{q}_j)} \\
&= \sqrt{\mathbf{Q}(\vec{q}_i - \vec{q}_j) \cdot \mathbf{Q}(\vec{q}_i - \vec{q}_j)} \\
&= \sqrt{\mathbf{Q}\mathbf{Q}^T(\vec{q}_i - \vec{q}_j) \cdot (\vec{q}_i - \vec{q}_j)} \\
&= \sqrt{(\vec{q}_i - \vec{q}_j) \cdot (\vec{q}_i - \vec{q}_j)} \\
&= q_{ij}
\end{aligned}
\tag{9}
$$

This implies that $V(\mathbf{Qq}) = V(\mathbf{q})$ and hence, $L(\mathbf{q}, \dot{\mathbf{q}}) = L(\mathbf{Qq}, \dot{\mathbf{q}})$. Thus, we demonstrate that the Lagrangian in MCLNN exhibits both translational and rotational symmetry. Invoking Noether's theorem, it can be proven that the Lagrangian preserving these symmetries will consequently conserve of linear and angular momenta, respectively. (For proof, see App. A.4 and A.5).

Combining Lemma 1 and Lemma 2, we get that the MCLNN will respect the laws of conservation energy, linear momentum, and angular momentum. □

### 3.4 GENERALIZABILITY AND INTERPRETABILITY

The MCLNN exhibits a granular structure, which applies the computations on individual or pairwise entities that are aggregated to obtain the total Lagrangian of the system. This structure, by design, allows model generalizability to unseen system sizes. Once trained on an $n$-body system, the MCLNN can be applied on an $m$-body system ($m \neq n$) directly by replacing $n$ with $m$ in Eq. 7.

Finally, we discuss interpretability of MCLNN. The main function learned by MCLNN is the $V_{ij}$, which depends on $q_{ij}$. Since, $V_{ij}$ adds to form the total potential energy $V(\mathbf{q})$ of the system, it is reasonable to assume that $V_{ij}$ represents the pair-wise potential energy. As we demonstrate later in Sec. 4.4, we show that $V_{ij}$ indeed corresponds to the pair-wise potential energy of the ground truth.

## 4 EXPERIMENTS

In this section, we benchmark MCLNN and establish:

- **Accuracy:** MCLNN is more accurate in modeling trajectories of $\mathcal{N}$-body systems than LNN (Cranmer et al., 2020a).
- **Conservation of Physics:** Consistent with the theoretical analysis in Sec. 3.3, MCLNN preserves the laws of physics more comprehensively than LNN
- **Generalizability and Interpretability:** MCLNN generalizes to unseen data better than LNN and enables higher interpretability due to predicting the potential energy instead of the Lagrangian directly.

The codebase along with all baselines used and the datasets are available at the anonymous repository https://anonymous.4open.science/r/nbodyMCLNN-2618/.

### 4.1 EXPERIMENTAL SETUP

We use JAX (Bradbury et al., 2020), a high-performance numerical computing python package and JAX MD (Schoenholz & Cubuk, 2020), a JAX based molecular dynamics package for the $n$-body simulations.

#### 4.1.1 TASKS

To benchmark the performance of MCLNN, we conduct experiments on three multi-body systems, namely, linear and non-linear springs, and gravitational systems.

• **Linear spring.** We simulate the dynamics of three particles (e.g.: balls with volume tending to zero) connected with each other by linear springs. The initial conditions of the simulations include the positions and velocities of the three particles and the stiffness of the spring. The potential energy of two balls connected by a spring is given by $V_{ij} = \frac{1}{2}k(q_{ij} - q_0)^2$, where $q_{ij}$ is the instantaneous distance between the springs and $q_0$ is the equilibrium distance, and $k$ is the stiffness of the spring. The force between connected pair of particles is computed as $F_{ij} = -\frac{dV_{ij}}{dq_{ij}} = -k(q_{ij} - q_0)$. Note that the value of $k$ is kept constant for all the springs.

• **Non-linear spring.** We simulate the dynamics of three particles connected by non-linear springs. The initial conditions of the simulations include the positions and velocities of the three particles. The potential energy of two particles connected by a non-linear spring is given by $V_{ij} = \frac{1}{2}k(q_{ij} - q_0)^4$, where $q_{ij}$ is the instantaneous distance between the springs and $q_0$ is the equilibrium distance, and $k$ is the stiffness of the spring. The force between connected pair of particles is computed as $F_{ij} = -2k(q_{ij} - q_0)^3$. The value of $k$ is kept constant for all the springs.

• **Gravitational system.** We simulate the dynamics of four particles interacting with each other under the gravitational force. The initial conditions of the simulations include the positions and velocities of the four masses. The potential energy of a pair of particles with mass $m_1$ and $m_2$ if given by $V_{ij} = -\frac{Gm_1m_2}{q_{ij}}$, where $G$ is universal gravitational constant and $q_{ij}$ is the instantaneous distance between the pair of particles. The force between the pair of particles is computed as $F_{ij} = \frac{Gm_1m_2}{q_{ij}^2}$. Note that the mass $m$ of all the particles are kept constant.

**Data Generation:** We use *forward simulation* of $\mathcal{N}$ particles to generate training data corresponding to each task. First, an $\mathcal{N}$-body configuration with given initial positions and velocities are considered. The potential energy of the system is computed from the positions using analytical expressions associated with each task as described above. The force on each particle is then calculated using as the gradient of potential energy, which is used to compute the acceleration. Finally, the *velocity-Verlet* algorithm is used for time integration to obtain the updated positions of the particles. Please see the appendix (A.6) for configurations (including the number of particles, initial configuration, and time step) of each of the tasks.

#### 4.1.2 BASELINE

The original LNN takes both the positions and velocities of all the particles as input and gives the total Lagrangian as the output. In the present case, we make the assumption on the functional form of kinetic energy (see Eq.5). As such, to ensure a fair comparison, a slightly modified version of the original LNN is considered here as the baseline. For this baseline, the positions are given as input to the neural network which gives the potential energy $V(\vec{q})$ as output. The kinetic energy is computed using the functional form, from which the Lagrangian is computed. All the remaining training procedures are maintained same as in the original LNN. It should be noted that decoupling the $T$ and $V$ terms and providing the expression for $T$ should make the learning easier for the LNN and hence should give improved performance in comparison to the original LNN. This model will be referred to as baseline or baseline LNN, henceforth.

#### 4.1.3 METRICS

To analyze the performance of the MCLNN in comparison to the baseline, the choice of the appropriate metrics is crucial. Since the systems considered here are chaotic, slight differences in the initial conditions or predictions will lead to differences in the trajectories. More importantly, given the current state of a system, multiple "correct" trajectories may exist as long as they represent the *degenerate* or equivalent configurational states of the system. As such, purely computing the error in trajectory with respect to the ground truth may not be representative of the performance of the model. Learning and predicting the Lagrangian ensures that the state of the system is represented ac-

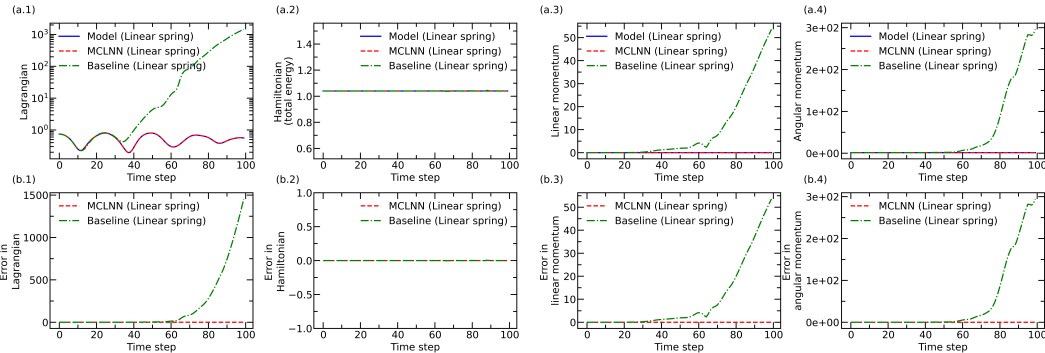

Figure 2: Results of the forward simulation of linear spring system from the same initial configuration using ground truth (continuous line), baseline LNN (dashed and dotted line) and MCLNN (dashed lines). (a) The evolution of Lagrangian, Hamiltonian (total energy), linear and momentum for all the systems. (b) The error in Lagrangian, Hamiltonian (total energy), linear and momentum predicted by the baseline LNN and MCLNN with respect to the ground truth.

curately, resulting in the prediction of the same or equivalent trajectories of the system. Further, the trajectories should respect the spatial and temporal symmetries resulting in conservation of Hamiltonian (total energy), linear, and angular momenta. Thus, we focus on the *Mean Absolute Error (MAE)* between predicted **(1)** Lagrangian, **(2)** Hamiltonian (total energy), **(3)** linear and **(4)** angular momenta, with respect to the ground truth.

### 4.1.4 TRAINING

For training MCLNN, 100 different trajectories, each having 20 points with fixed time intervals, are used for all three tasks. Each trajectory has different initial conditions and hence represent different configurations that are accessible to the system. The model performance is evaluated by comparing the predicted trajectory with respect to the ground truth for the same initial state.

For training LNN, we sample 10,000 data points (set of $(\mathbf{q}, \dot{\mathbf{q}})$ as input) and $\ddot{\mathbf{q}}$ as output) from forward simulations. The training is performed using the mean squared error between the acceleration predicted by the EL equation and the ground truth. The detailed configurations and parameters associated with simulation and training of each task is provided in App. A.6.

### 4.1.5 INFERENCE

To asses the long term stability of models, we perform forward simulations using trained LNN and MCLNN models for a prolonged time interval for each task. The initial condition and other parameters are kept same for both LNN and MCLNN. Furthermore, the initial conditions of the trajectories during inference are different from those encountered during the training. Consequently, all trajectories during evaluation are *unseen*. We record Lagrangian, Hamiltonian, linear momentum and angular momentum for the whole trajectory. The predicted forward simulation is compared with the ground truth data as generated in Sec. 4.1.1.

### 4.2 ACCURACY

First, we focus on the linear spring system with three particles. Figure 2(a.1)-(a.4) shows the Lagrangian, Hamiltonian (total energy), linear, and angular momenta of the system. The error in these quantities predicted by the baseline LNN and MCLNN are shown in Figure 2(b.1)-(b.4). We observe that the Lagrangian predicted by the baseline LNN starts diverging after 30 time steps of the forward simulation. We notice that the divergence in the Lagrangian is accompanied by a divergence in the linear momentum as well. Up on further simulation, the angular momentum also starts diverging. In contrast, the Lagrangian predicted by the MCLNN follows the ground truth with very little error. Further, the Hamiltonian, and linear and angular momenta remain conserved in the MCLNN. This suggests that the trajectory predicted by the MCLNN is accurate and stable with no long term drift or divergence. It is worth noting that, for the baseline LNN, despite the poor predictions of the Lagrangian at higher values of time steps, the total energy remain conserved. This points to the fact that the energy conservation is a constraint enforced by the EL equation and is not a reflection on the quality Lagrangian. Any value of Lagrangian can still yield a constant Hamiltonian (energy), provided the constraints as imposed by the EL equations on the Lagrangian are satisfied. As such, LNNs should be evaluated considering multiple metrics as demonstrated here.

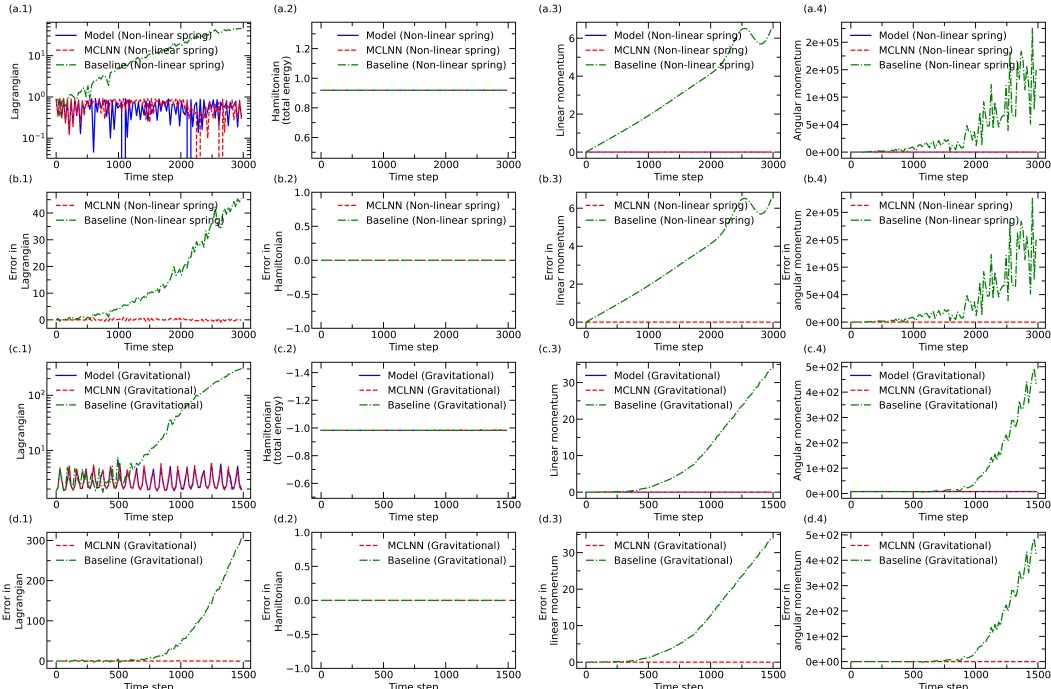

Figure 3: Results of the forward simulation of non-linear spring, and gravitational system from the same initial configuration using ground truth (continuous line) and MCLNN (dashed lines). (a) The evolution of Lagrangian, Hamiltonian (total energy), linear and momentum for all the systems. (b) The error in Lagrangian, Hamiltonian (total energy), linear and momentum predicted by the baseline LNN and MCLNN with respect to the ground truth.

Now, we focus on the non-linear spring and the gravitational system. Figure 3(a) and (c) shows the Lagrangian, Hamiltonian (total energy), linear, and angular momenta of the non-linear spring and gravitational systems. The error in these quantities predicted by the baseline LNN and MCLNN are shown in Figure 2(b) and (d). As in the case of the linear spring system, we observe that there is a long term drift in the Lagrangian predicted by the baseline LNN. Similarly, the linear and angular momenta of the baseline also diverge in a few time steps. In contrast, the Lagrangian predicted by the MCLNN exhibits a good match with the ground truth. It should be noted that the multi-particle interacting systems are chaotic in nature and hence the exact trajectory of the ground truth and simulated system may vary. However, the low values of error in the Lagrangian along with the long-term stability of both non-linear spring and gravitational system suggests the realistic nature of the trajectory predicted by MCLNN. In addition, we observe that the linear and angular momenta of the system remain conserved in the MCLNN. Altogether, the results confirm that the MCLNN can successfully learn the generative model $p(\mathbb{T})$ that can sample the trajectory effectively from the physics-constrained configurational and temporal space.

### 4.3 GENERALIZABILITY TO UNSEEN SYSTEM SIZES

To demonstrate the generalizability of MCLNN, we use the generative model $p(\mathbb{T})$ for each of the tasks learned during the training. The models are then used to predict trajectories of systems

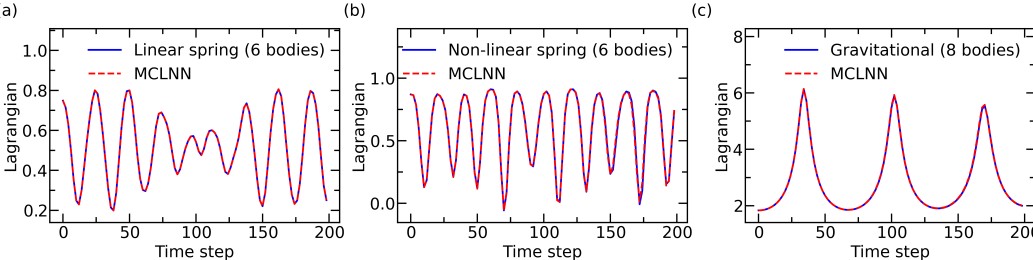

Figure 4: Results of the forward simulation of (a) linear spring with 6 particles, (b) non-linear spring with 6 particles, and (b) gravitational system with 8 particles from the same initial configuration using ground truth (continuous line) and MCLNN (dashed lines). The evolution of Lagrangian for all the systems are plotted.

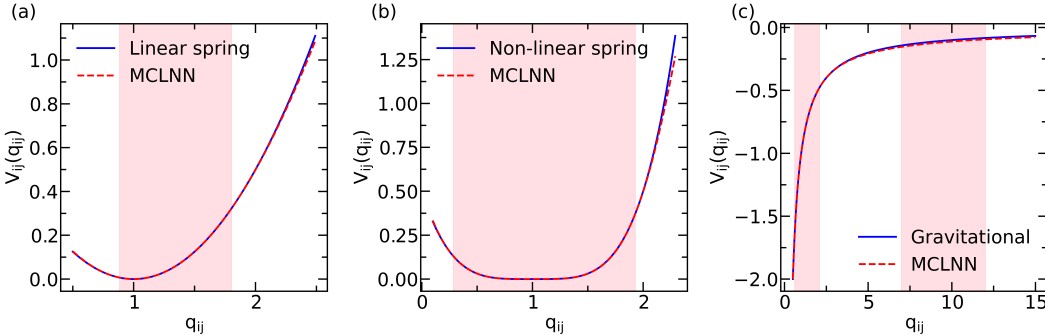

Figure 5: Variation of $V_{ij}$ with respect to $q_{ij}$ as learned by the MCLNN (dashed line) in comparison to the ground truth (continuous line) for (a) linear spring, (b) non-linear spring, and (c) gravitational potential. The shaded region represents the range of $q_{ij}$ values in the training set. Non-shaded region between two shaded region corresponds to interpolation; otherwise, it represents extrapolation.

with different number of particles. Figure 4 shows the trajectories for linear and non-linear springs with six particles (MCLNN trained on three particles), and gravitational system with eight particles (MCLNN trained on four particles), predicted by the MCLNN in comparison to the ground truth. We observe that the Lagrangian predicted by the MCLNN exhibits excellent match with the ground truth. In addition, all other quantities such as energy and momenta remain conserved (see Appendix Fig. 7). This suggests that once the generative model $p(\mathbb{T})$ has been learned, MCLNN can generalize it to predict the trajectory of systems with any number of particles. Note that the baseline LNN cannot simulate systems of different sizes as the positions of all the particles are given simultaneously as the input to the neural network.

### 4.4 INTERPRETABILITY

Figure 5 shows the $V_{ij}$ predicted by the MCLNN in comparison to the pair-wise potential energy obtained analytically from ground truth. The shaded region represents the $q_{ij}$ values that were present in the training set. We observe that the MCLNN is able to learn the pairwise potential energy function accurately for the $q_{ij}$ values in the training set. Further, the MCLNN is able to interpolate values excellently and extrapolate reasonably. It is worth noting that the MCLNN is trained only on the trajectories of the system. Hence, no information regarding the potential energy, force, or even acceleration is given to the neural network during training. This suggests that MCLNN is indeed able to learn the physics of the problem by training purely on the trajectory.

## 5 CONCLUSION

We introduced a new framework, namely, *momentum conserving Lagrangian neural networks* (MCLNN), for incorporating physics-based priors in neural networks for accurate simulations of multi-particle systems. We demonstrated that the MCLNN respects the symmetries in space and time, leading to conservation of energy, and linear and angular momentum. We showed that the incorporation of these additional conservation laws of momenta makes the Lagrangian of the system stable by avoiding any long-term drift in it. This, in turn, results in a realistic simulation of multi-body systems. Further, we showed that the MCLNN once trained, can generalize to systems of any size. Finally, we demonstrated that the MCLNN is highly interpretable and provides direct insights into the interaction laws governing the dynamics of multi-particle systems. This, in turn, allows one to verify the realistic nature of the function learned by the MCLNN.

At this juncture, it is worth discussing some of the open questions and shortcomings of MCLNN that can be addressed as part of future works. **(i)** MCLNN assumes interaction between all the particles in the system. A graph-based MCLNN with contrastive loss can potentially address this challenge by incorporating the topology of the particle system along with differential importance for nearby particles. **(ii)** MCLNN considers only pair-wise interactions for computing the $V_{ij}$. However, there could be additional interactions involving three ($V_{ijk}$), four ($V_{ijkl}$) or even higher particles. This could be addressed by feature engineering or graph-based MCLNN. **(iii)** Similarly, simulations involving different types of interacting particles is challenging in MCLNN as it may lead to non-unique solutions. **(iv)** In addition, present work can be extended to address more challenging problems, for instance, to learn the Lagrangian of system with non-conservative forces, and to learn generalized kinetic energy functions, while maintaining the granularity and generalizability.

## 6 Reproducibility Statement

Please find all codes and experiments at the following anonymous link: https://anonymous.4open.science/r/nbodyMCLNN-2618/. In Appendix A.6, we have given additional details to make the work completely reproducible. In particular, we have: (1) additional derivations and proofs in App. A.3,A.4,A.5, (2) initial conditions, task configurations, and parameters for the training of neural networks used for each task in App. A.6, (3) detailed notation in App. A.7. The dataset generation (using simulations) step is added to the code before training step and hence there are no additional dataset files are provided.

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

# A APPENDIX

## A.1 BASELINE LNN

Figure 6 shows the predicted force with respect to the actual force for the training data of baseline for the three tasks.

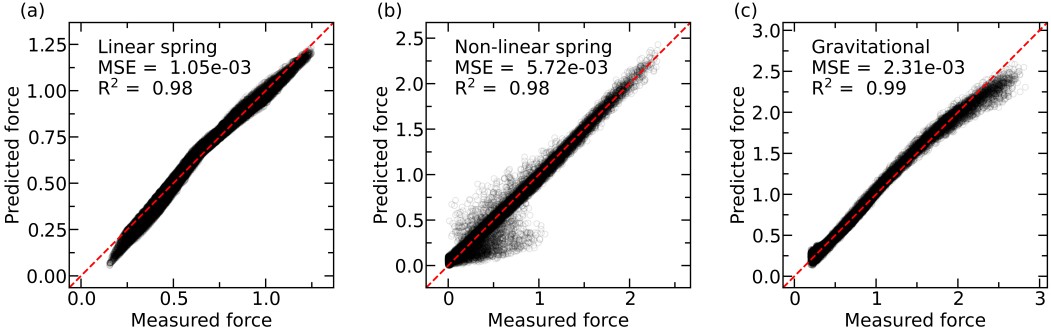

Figure 6: Predicted force for (a) linear spring, (b) non-linear spring, and (c) gravitational system with respect to the measured forces on the test dataset of baseline LNN.

## A.2 GENERALIZABILITY TO UNSEEN SYSTEM SIZES

Figure 7 shows the generalizability of the MCLNN to unseen system sizes for the three tasks considered. For all the three tasks, the number of particles considered are twice that of the training system size. We observe that the Lagrangian for the unseen system is predicted in excellent agreement to the ground truth for all the three tasks. In addition, the Hamiltonian (total energy), and momenta are conserved in all the three tasks, confirming the realistic nature of the simulations by MCLNN.

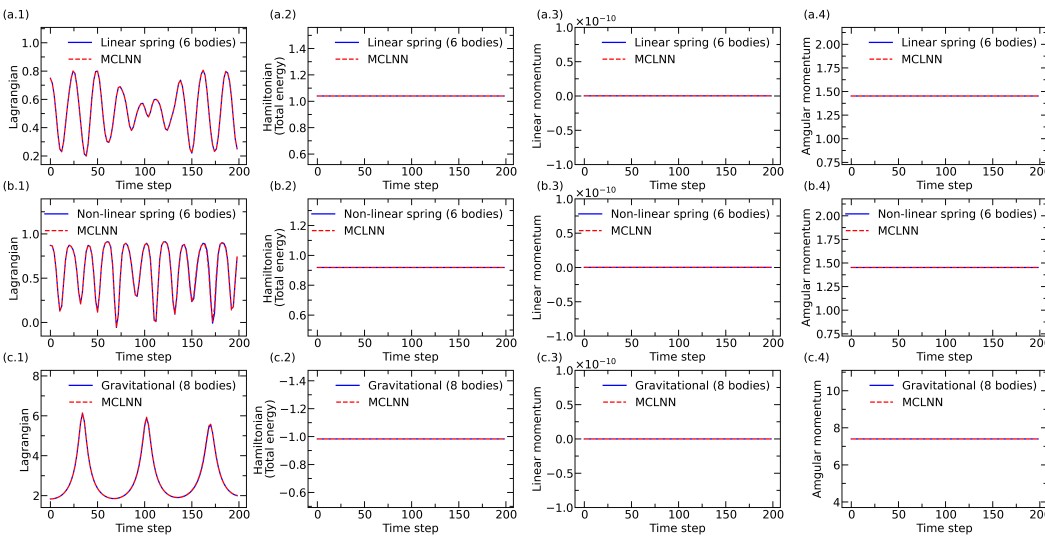

Figure 7: Results of the forward simulation of (a) linear spring, (b) non-linear spring, and (b) gravitational system from the same initial configuration using ground truth (continuous line) and MCLNN (dashed lines). The evolution of Lagrangian, Hamiltonian (total energy), linear and angular momenta for all the systems with different number of particles than that was present in the training set are plotted.

## A.3 Computing Acceleration from Euler-Lagrange Equations

The acceleration of particles can be obtained from the Euler-Lagrange equations as follows.

$$\frac{\partial}{\partial \dot{\mathbf{q}}} \frac{dL}{dt} = \frac{\partial L}{\partial \mathbf{q}}$$

$$\frac{\partial}{\partial \dot{\mathbf{q}}} \left( \frac{\partial L}{\partial \mathbf{q}} \frac{dq}{dt} + \frac{\partial L}{\partial \dot{\mathbf{q}}} \frac{d\dot{\mathbf{q}}}{dt} \right) = \frac{\partial L}{\partial \mathbf{q}}$$

$$\frac{\partial}{\partial \dot{\mathbf{q}}} \left( \frac{\partial L}{\partial \mathbf{q}} \dot{\mathbf{q}} + \frac{\partial L}{\partial \dot{\mathbf{q}}} \ddot{\mathbf{q}} \right) = \frac{\partial L}{\partial \mathbf{q}}$$

$$\frac{\partial}{\partial \dot{\mathbf{q}}} \frac{\partial L}{\partial \mathbf{q}} \dot{\mathbf{q}} + \frac{\partial}{\partial \dot{\mathbf{q}}} \frac{\partial L}{\partial \dot{\mathbf{q}}} \ddot{\mathbf{q}} = \frac{\partial L}{\partial \mathbf{q}}$$

$$\frac{\partial}{\partial \dot{\mathbf{q}}} \frac{\partial L}{\partial \dot{\mathbf{q}}} \ddot{\mathbf{q}} = \frac{\partial L}{\partial \mathbf{q}} - \frac{\partial}{\partial \dot{\mathbf{q}}} \frac{\partial L}{\partial \mathbf{q}} \dot{\mathbf{q}}$$

$$\ddot{\mathbf{q}} = \left( \frac{\partial}{\partial \dot{\mathbf{q}}} \frac{\partial L}{\partial \dot{\mathbf{q}}} \right)^{-1} \left[ \frac{\partial L}{\partial \mathbf{q}} - \frac{\partial}{\partial \dot{\mathbf{q}}} \frac{\partial L}{\partial \mathbf{q}} \dot{\mathbf{q}} \right]$$

$$\ddot{\mathbf{q}} = \left( \nabla_{\dot{\mathbf{q}}\dot{\mathbf{q}}} L \right)^{-1} \left[ \nabla_q L - \left( \nabla_{\dot{\mathbf{q}}\mathbf{q}} L \right) \dot{\mathbf{q}} \right]$$

## A.4 Translational Symmetry

The conservation of linear momentum can be derived from translational symmetry of Lagrangian as follows.

$$L(\mathbf{q}, \dot{\mathbf{q}}) = L(\mathbf{q} + \epsilon, \dot{\mathbf{q}})$$

$$\mathbf{q}' = \mathbf{q} + \epsilon$$

$$\implies \frac{\partial \mathbf{q}'}{\partial \mathbf{q}} = 1$$

$$\sum \frac{\partial L}{\partial \mathbf{q}} = \sum \frac{\partial L}{\partial \mathbf{q}'} = \sum \frac{\partial L}{\partial \mathbf{q}} \frac{\partial \mathbf{q}}{\partial \mathbf{q}'} = \sum \frac{\partial L}{\partial \mathbf{q}}.1$$

$$\delta L = \sum \delta \mathbf{q} \frac{\partial L}{\partial \mathbf{q}}$$

$$\delta L = \sum \epsilon \frac{\partial L}{\partial \mathbf{q}}$$

$$\delta L = \epsilon \sum \frac{\partial L}{\partial \mathbf{q}}$$

since,

$$\delta L = 0$$

$$\implies \sum \frac{\partial L}{\partial \mathbf{q}} = 0$$

$$\sum \frac{\partial L}{\partial \mathbf{q}} = 0$$

$$\sum \frac{d}{dt} \frac{\partial L}{\partial \dot{\mathbf{q}}} = 0$$

$$\frac{d}{dt} \sum \frac{\partial L}{\partial \dot{\mathbf{q}}} = 0$$

$$\sum \frac{\partial L}{\partial \dot{\mathbf{q}}} = constant$$

$$\sum p_i = \text{constant (linear momentum)}$$

where $p_i = \frac{\partial L}{\partial \dot{q}_i}$

## A.5 ROTATIONAL SYMMETRY

The conservation of angular momentum can be derived from rotational symmetry of Lagrangian as follows.

Under an infinitesimal rotation $\delta\theta$

$$\delta\vec{q}_i = \delta\theta \times \vec{q}_i$$

and

$$\delta\dot{\vec{q}}_i = \delta\theta \times \dot{\vec{q}}_i$$

since, Lagrangian does not change with infinitesimal rotation $\delta\theta$

$$\delta L = \sum_i \frac{\partial L}{\partial \vec{q}_i} \delta\vec{q}_i + \sum_i \frac{\partial L}{\partial \dot{\vec{q}}_i} \delta\dot{\vec{q}}_i = 0$$

using generalized momentum

$$\frac{\partial L}{\partial \dot{\vec{q}}_i} = \vec{p}_i$$

then EL equation gives

$$\frac{d}{dt}\vec{p}_i - \frac{\partial L}{\partial \vec{q}_i} = 0$$

$$\dot{\vec{p}}_i = \frac{\partial L}{\partial \vec{q}_i}$$

$$\implies \delta L = \sum_i \dot{\vec{p}}_i \cdot \delta \vec{q}_i + \sum_i \vec{p}_i \cdot \delta \dot{\vec{q}}_i = 0$$

$$\sum_i \dot{\vec{p}}_i \cdot (\delta\theta \times \vec{q}_i) + \vec{p}_i \cdot (\delta\theta \times \dot{\vec{q}}_i) = 0$$

$$\sum_i \delta\theta \cdot (\vec{q}_i \times \dot{\vec{p}}_i) + \delta\theta \cdot (\dot{\vec{q}}_i \times \vec{p}_i) = 0$$

$$\sum_i \delta\theta \cdot [(\vec{q}_i \times \dot{\vec{p}}_i) + (\dot{\vec{q}}_i \times \vec{p}_i)] = 0$$

$$\sum_i \delta\theta \cdot \frac{d}{dt}(\vec{q}_i \times \vec{p}_i) = 0$$

because $\delta\theta$ is arbitrary, then

$$\sum_i \frac{d}{dt}(\vec{r}_i \times \mathbf{p}_i) = 0$$

$$\frac{d}{dt}(\sum_i \vec{r}_i \times \mathbf{p}_i) = 0$$

$$\text{angular momentum} = \sum_i (\vec{r}_i \times \vec{p}_i) = \text{constant}$$

## A.6    Initial Conditions and Task Configuration

The number of hidden units are chosen from hyperparamter search given in Table 1. The activation function for all hidden units are square plus $f(x) = \frac{(x+\sqrt{x^2+4})}{2}$ which is similar to soft plus (Cranmer et al., 2020a). We use ADAM (Kingma & Ba, 2015) optimiser for model training with learning rate given in respective configuration tables. We use velocity-Verlet for time integration during trajectory evolution. Mass of all the particles are maintained as 1.0 units.

### A.6.1    Linear Spring System

$$\text{Initial position} = \begin{bmatrix} 0.486657678894505 & 0.755041888583519 & 0.0 \\ -0.681737994414464 & 0.293660233197210 & 0.0 \\ -0.022596327468640 & -0.612645601255358 & 0.0 \end{bmatrix}$$

$$\text{Initial velocity} = \begin{bmatrix} -0.182709864466916 & 0.363013287999004 & 0.0 \\ -0.579074922540872 & -0.748157481446087 & 0.0 \\ 0.761784787007641 & 0.385144193447218 & 0.0 \end{bmatrix}$$

Table 1: Train and validation loss for each NN architecture for the linear spring task for MCLNN. Note that for hyperparamter search (i.e. number of hidden units), minimum train loss was set to 1.0e-8 as stopping criteria.

| Hidden Layers | Linear Spring |
| --- | --- |
| 2, 2 | 0.0012 , 0.0015 |
| 4, 4 | 7.98e-09, 2.36e-08 |
| 8, 8 | 9.86e-09, 3.80e-08 |
| 16, 16 | 9.76e-09, 3.02e-08 |

dt $= 0.01$
mass $= 1.0$
stride $= 10$
runs $= 20 =$ points per trajectory
lr $= 1.0e - 3$
layers $= [10, 10]$
epochs $= 100000$
samples $= 100 =$ number of trajectories
seed $= 100$
time step $= dt \times stride = 0.1$

### A.6.2 NON-LINEAR SPRING SYSTEM

$$\text{Initial position} = \begin{bmatrix} 0.486657678894505 & 0.755041888583519 & 0.0 \\ -0.681737994414464 & 0.293660233197210 & 0.0 \\ -0.022596327468640 & -0.612645601255358 & 0.0 \end{bmatrix}$$

$$\text{Initial velocity} = \begin{bmatrix} -0.182709864466916 & 0.363013287999004 & 0.0 \\ -0.579074922540872 & -0.748157481446087 & 0.0 \\ 0.761784787007641 & 0.385144193447218 & 0.0 \end{bmatrix}$$

dt $= 0.01$
mass $= 1.0$
stride $= 10$
runs $= 20 =$ points per trajectory
lr $= 1.0e - 3$
layers $= [10, 10]$
epochs $= 100000$
samples $= 100 =$ number of trajectories
seed $= 100$
time step $= dt \times stride = 0.1$

### A.6.3 GRAVITATIONAL SYSTEM

$$\text{Initial position} = \begin{bmatrix} 1.0 & 0.0 & 0.0 \\ 9.0 & 0.0 & 0.0 \\ 11.0 & 0.0 & 0.0 \\ -1.0 & 0.0 & 0.0 \end{bmatrix}$$

$$\text{Initial velocity} = \begin{bmatrix} 0.0 & 0.05 & 0.0 \\ 0.0 & -0.05 & 0.0 \\ 0.0 & 0.65 & 0.0 \\ 0.0 & -0.65 & 0.0 \end{bmatrix}$$

dt $= 0.01$
mass $= 1.0$
stride $= 10$
runs $= 20 =$ points per trajectory
lr $= 1.0e - 3$
layers $= [10, 10]$
epochs $= 100000$
samples $= 100 =$ number of trajectories
seed $= 100$
time step $= dt \times stride = 0.1$

## A.7 DEFAULT NOTATION

| | |
|---|---|
| $a$ | A scalar (integer or real) |
| $\vec{a}$ | A vector |
| $\mathbf{A}$ | A matrix |
| $\boldsymbol{I}_n$ | Identity matrix with $n$ rows and $n$ columns |
| $\boldsymbol{I}$ | Identity matrix with dimensionality implied by context |
| $\boldsymbol{e}^{(i)}$ | Standard basis vector $[0, \ldots, 0, 1, 0, \ldots, 0]$ with a 1 at position $i$ |
| $\vec{a}_i$ | Vector corresponding to particle $i$ |
| $A_{i,j}$ | Element $i, j$ of matrix $\boldsymbol{A}$ |
| $\boldsymbol{A}_{i,:}$ | Row $i$ of matrix $\boldsymbol{A}$ |
| $\boldsymbol{A}_{:,i}$ | Column $i$ of matrix $\boldsymbol{A}$ |
| $\boldsymbol{A}_{i,j,k}$ | Element $(i, j, k)$ of a 3-D tensor $\mathbf{A}$ |
| $\mathbf{A}_{:,:,i}$ | 2-D slice of a 3-D tensor |
| $\dfrac{dy}{dx}$ | Derivative of $y$ with respect to $x$ |
| $\dfrac{\partial y}{\partial x}$ | Partial derivative of $y$ with respect to $x$ |
| $\nabla_{\vec{x}} L$ | Gradient of $L$ with respect to $\vec{x}$ |
| $\nabla_{\boldsymbol{X}} y$ | Matrix derivatives of $y$ with respect to $\boldsymbol{X}$ |
| $\dfrac{\partial f}{\partial \boldsymbol{x}}$ | Jacobian matrix $\boldsymbol{J} \in \mathbb{R}^{m \times n}$ of $f : \mathbb{R}^n \to \mathbb{R}^m$ |
| $\displaystyle\int f(\boldsymbol{x}) d\boldsymbol{x}$ | Definite integral over the entire domain of $\boldsymbol{x}$ |
| $\displaystyle\int_{\mathbb{S}} f(\boldsymbol{x}) d\boldsymbol{x}$ | Definite integral with respect to $\boldsymbol{x}$ over the set $\mathbb{S}$ |
| $P(\mathrm{a})$ | A probability distribution over a discrete variable |
| $p(\mathrm{a})$ | A probability distribution over a continuous variable, or over a variable whose type has not been specified |

