# OpenReview forum: "Momentum Conserving Lagrangian Neural Networks"
_ICLR.cc/2022/Conference — ICLR 2022 Submitted_

### Official Review · Reviewer_Qsuh · 2021-10-24

**Correctness:** 4
**Technical Novelty And Significance:** 1
**Empirical Novelty And Significance:** 2
**Recommendation:** 3
**Confidence:** 4

**Main Review:**

**Pros**

The paper is well-organized and self-explanatory. The theoretical backgrounds such as Noether's theorem and the momentum conservation law are introduced in detail and in a reader-friendly way.

The performance improvement is valid by reliable experiments and supported by theories.

**Cons**

My main concern is about novelty and significance.

For multi-body problems, the potential energy is the center of attention, while the kinetic energy is easily modeled. Among the community of molecular dynamics, an empirical approximation of the potential energy has a long history. One of the most famous ones is the Lennard-Jones potential [1]. Many studies have attempted to approximate the potential function by using machine learning methods, and many recent studies have employed neural networks [2][3][4]. The improvement from these studies is unclear.

- [1] J. E. Jones (1924). "On the determination of molecular fields.—I. From the variation of the viscosity of a gas with temperature". Proceedings of the Royal Society of London. Series A, Containing Papers of a Mathematical and Physical Character.
- [2] E. D. Cubuk, B. D. Malone, B. Onat, A. Waterland, and E. Kaxiras (2017) "Representations in neural network based empirical potentials". Journal of Chemical Physics.
- [3] M. Wen and E. B. Tadmor. (2019) "Hybrid neural network potential for multilayer graphene". Physical Review B.
- [4] J. Han, L. Zhang, R. Car, and W. E (2020). "Deep Potential: A General Representation of a Many-Body Potential Energy Surface". Communications in Computational Physics.

In my opinion, if a theory well-known in another community contributes greatly to the neural network community, an introduction of the theory should be valued highly. However, the contribution of the present study is directed to the computational science community rather than the neural network community. Hence, the evaluation from the viewpoint of the computational science is important. In this sense, the present study does not demonstrate a sufficient novelty or significance.

**Other Comment**

The proposed method models the potential function as the sum of pair-wise potential functions. The proposed method is a special case of DeLaN [5], which approximates the whole potential function using a neural network. DeLaN is also a special case of LNN [6], which approximates the Lagrangian. Related works should be introduced in more detail.

The proposed method is useful but sometimes insufficient. The potential function of a planatry system (a.k.a. gravitational system) is one of the targets of the proposed method, but such system is too simple to be modeled by neural networks. The potential function of a molecular dynamics is led by the nonlinear interaction among more than two particles described by the Schrödinger equation. The proposed method is applicable only to the cases where the nonlinearity is negligible.

- [5] M. Lutter, C. Ritter, and J. Peters (2019). "Deep Lagrangian Networks: Using Physics as Model Prior for Deep Learning". ICLR.
- [6] M. Cranmer, S. Greydanus, S. Hoyer, P. Battaglia, D. Spergel, and S. Ho. (2020) "Lagrangian neural networks". ICLR 2020 Workshop on Integration of Deep Neural Models and Differential Equations.


**Summary Of The Paper:**

Many recent studies have attempted to learn Lagrangian and Hamiltonian systems by using neural networks, but most of them still suffer from instability to learn the translational and rotational symmetries. These symmetries are associated with the conservation laws of linear momentum and angular momentum, respectively. In contrast to Lagrangian neural network, which learns a Lagrangian as a black-box, the proposed method learns the Lagrangian as the kinetic energy predefined by (5) minus the sum of pair-wise potential energy $V_{ij}$ modeled by a neural network. Thanks to this explicit formulation, the proposed method conserves not only the system energy but also linear/angular momentum, as confirmed by the experiments.

**Summary Of The Review:**

The manuscript is well-organized, and the experiments are reliable. The main concern is the novelty. Many similar approaches have been investigated in the community of molecular dynamics, and the improvement from them is unclear.

---

### Official Review · Reviewer_1UZJ · 2021-11-01

**Correctness:** 4
**Technical Novelty And Significance:** 2
**Empirical Novelty And Significance:** 2
**Recommendation:** 3
**Confidence:** 4

**Main Review:**

Major concerns

- It seems to me that the kinetic energy is not learned. Instead the ground truth is used during training. This model design is not in the spirit of HNN and LNN line of works, where masses are assumed unknown and need to be learned. If only potential energy is learned, I think a name like "potential energy net" is more appropriate than MCLNN. What if the authors assume that masses are not known and instead need to be learned? You can assume the form of kinetic energy (eqn. 5) is known. Then the question is under this setting, how does the model perform in terms of momentum conservation? You can even comment on the learned and true masses. I think this is more reasonable setting if you want to enforce momentum conservation into LNN.
- The description of the generative model to "generalize to unseen system sizes" is missing. Since in all the systems, the pairwise interactions is homogeneous, I guess the neural network is learning the pairwise potential energy. In other words, since the spring constants and the masses (in the gravitational experiments) are the same (for each pair). The neural network can learn the pairwise potential energy and they are summed up to get the total potential energy. In this way, the learned model can be easily generalized to unseen system sizes. If this is true, this kind of generalization cannot be done with heterogeneous pairwise interactions, e.g., different spring constants in a system. This is a big limitation on the claim of "generalization to unseen system sizes" and should be pointed out clearly in the paper.

Minor concerns
- Why did you choose square plus to be the activation function? Did you encounter any problem using softplus?
- In section 4.2, the authors says
>Any value of Lagrangian can still yield a constant Hamiltonian (energy),provided the constraints as imposed by the EL equations on the Lagrangian are satisfied. As such, LNNs should be evaluated considering multiple metrics as demonstrated here

I do not agree with this statement. If we compare an MLP, a LNN and a MCLNN. We know that an MLP has no physics incorporated, so none of the conservation quantities in the prediction trajectory should be expected. A LNN enforces energy conservation, so energy conservation is expected. A MCLNN enforces both energy conservation and momentum conservation, so these convervations are expected in the prediction trajectory. LNN and MCLNN are different models that target different problems. It is not fair to expect LNN to conserve momentum simply because it is not enforced. In other words, consider a hypothetical case: another work that extends MCLNN in the future and says "MCLNN should be evaluated on metrics other than energy and momentum conservations and you see, MCLNN perform badly on these additional metrics." This won't be fair to MCLNN as well. MCLNN conserves energy and momentum simply because it is designed to do so. We should not expect MCLNN to perform well on another, if any, metric that it is not enforced to do.
- Typo
    - section 4.1.5 asses -> assess
    - Figure 4 amd Figure 7 caption (a), (b), (b) -> (a), (b), (c)

**Summary Of The Paper:**

This paper is motivated by LNN and aim to enforce momentum conservation in learning based on the Lagrangian framework. The goal is straightforward and simple, but I have not seen similar work in the literature. The kinetic energy is assumed to be known and pairwise potential energy is learned through a neural network. Since the distance between each pair of particles is constructed as the input to the potential energy network, the learned model enforces energy as well as momentum conservation. The proposed model is tested on three Interacting particle systems with homogeneous pairwise interaction.

**Summary Of The Review:**

The paper is well written and the idea is simple and straightforward. The novelty is limited. My main concern is about the design of the known kinetic energy (mass) and the claim of generalizability to unseen system sizes. The known kinetic energy design does not make sense. The generalizability to unseen system sizes is only valid for homogeneous pairwise potential energy. The whole framework seems to only work for a specific system - interacting particles. This limited applicability also weakens the contribution. I think this submission does not meet the ICLR standard, thus I recommend rejection.

To meet the standard of major AI conferences, the paper should at least learn masses simultaneously and explore more application scenarios other than interacting particles.

---

### Official Review · Reviewer_Vgjt · 2021-11-02

**Correctness:** 4
**Technical Novelty And Significance:** 2
**Empirical Novelty And Significance:** 2
**Recommendation:** 3
**Confidence:** 5

**Main Review:**

The proposed modification of adding in the kinetic energy and only learning a parametrized potential is principled, well explained and one can see why it should work better, both in terms of modelling accuracy and physical soundness.

However, I find the scope of the paper to be too narrow. It is only a marginal transformation of an existing model, the theoretical contributions are quite elementary and the experiments do not really suggest wider applicability of the model in practice. I think this work is more suited to be a (very good) workshop submission.

**Summary Of The Paper:**

By constraining the form of the lagrangian and predicting only the value of the potential of a system of interacting particles, this paper proposes an improved version of the Lagrangian Neural Networks model which is shown to conserve the momentum of the system.

**Summary Of The Review:**

While technically sound, I don't advocate for this work to be accepted at ICLR as the contribution is too marginal.

---

### Official Review · Reviewer_N3Bq · 2021-11-04

**Correctness:** 4
**Technical Novelty And Significance:** 2
**Empirical Novelty And Significance:** 2
**Recommendation:** 5
**Confidence:** 3

**Main Review:**

## Pros:
- The empirical evaluation demonstrates a real gain of taking into account the conservation of additional quantities. It is indeed a good inductive bias to put in the model when we can do it.
- The simplicity of the idea is a plus and it is clearly introduced.

## Cons
- Although I found the manuscript overall easy to follow and clear, I think the presentation could be improved. As specific examples, the many bullet points in the introduction breaks the flow. Same with "Problem 1", actually I did not get what was the purpose and formulating the learning problem like that as it is not really in the rest of the manuscript except for the loss. I find the usage of the word "generative model" a bit confusing in the context of a physical model (although I agree those are valid generative models).
-  Lemma 1 and 2 are quite straightforward and so their proofs could go in the appendix I think.
- I am surprised nobody tried to use GNNs as Lagrangian neural networks as this seems quite natural for multi bodied systems. Actually, you also mentioned this in your conclusion and I wonder how the current paper is too incremental.
- As said in the previous point I think the contribution is very incremental.
- Could you clarify equation 6 and how you backprop through p(T q)?
- It is not clear if (4) is expressive enough for all kinds of systems, I suppose some could require higher-level interaction scheme for evaluating the potential energy?!
-
## Additional remarks
- 3.1 "then trajectory" missing determinant.
- just before (5) missing "be"
- The L when substituted... eq 2 => should be (3) right?
- page 7 up on -> upon


**Summary Of The Paper:**

This paper enhances Lagrangian neural networks by adding conservation of the angular and linear momenta. It does so by enforcing symmetry with respect to translation and rotation of the system in the Lagrangian. Experiments performed on a set of physical systems shows angular and linear momenta conservation reduce the divergence between the (implicitly) learned and exact dynamic of those systems.

**Summary Of The Review:**

Incremental but sound idea. Presentation could be improved.

---

### Decision · Program_Chairs · 2022-01-20

**Decision:**

Reject

**Comment:**

This paper enhances Lagrangian neural networks by adding conservation of the angular and linear momenta. According to the reviewers, the technical contribution of the paper is marginal, it is a incremental change of an existing model, and it seems that there is some over claim on the generalization of the model to unseen systems. The theoretical contributions in the paper are not significant, and the experiments have not demonstrate the practical potential of the proposed model yet. After the reviewers provided their comments, the authors did not submit their rebuttals. Therefore, as a result, we do not think the paper is ready for publication at ICLR.